



# 1  Improvement in algorithms for quality control of weather radar data
# 2  (RADVOL-QC system)

Katarzyna Ośródka[1], Jan Szturc[1]
[1]Centre of Meteorological Modelling, Institute of Meteorology and Water Management – National Research Institute, PL 01-
673 Warszawa, ul. Podleśna 61, Poland
*Correspondence to*: Katarzyna Ośródka (katarzyna.osrodka@imgw.pl)
**Abstract.** Data from weather radars are commonly used in meteorology and hydrology, but they are burdened with serious
disturbances, especially due to the appearance of numerous non-meteorological echoes. For this reason, these data are
subject to advanced quality control algorithms. The paper presents a significant improvement of the RADVOL-QC system
made necessary by the appearance of an increasing number of various disturbances. New algorithms are mainly addressed to
the occurrence of clutter caused by wind turbines (DP.TURBINE algorithm) and other terrain obstacles (DP.NMET
algorithm), as well as various forms of echoes caused by the interaction of a radar beam with RLAN signals (set of SPIKE
algorithms). The individual algorithms are based on the employment of polarimetric data as well as on the geometric
analysis of echo patterns. In the paper the algorithms are described along with examples of their performance and an
assessment of their effectiveness, and finally examples of the performance of the whole system are discussed.
**1. Introduction**
Weather radar data are widely employed in weather monitoring and forecasting, thus they have been significantly improved,
and observational capabilities of radars have been enhanced in response to increasing demands for better resolution and
accuracy. In particular, the enhancement of the accuracy of a quantitative precipitation estimation (QPE) has been a primary
goal in weather radar applications in meteorology and hydrology. Moreover, the upgrade to dual-polarization has promised
substantial improvements in data reliability since polarimetric parameters offer many opportunities by compensating for
various factors that cause measurement errors, such as the occurrence of different types of non-meteorological echoes,
especially those caused by interfering radio local area network (RLAN) signals and wind turbines (e.g., Bringi et al., 2011;
Saltikoff et al., 2019).

Radar precipitation estimates are affected by numerous sources of uncertainty (e.g. Villarini and Krajewski, 2010)
and require advanced post-processing algorithms. While employing weather radar observations it is crucial to perform an
advanced quality control (QC) of the data, which consists of clearing it from erroneous echoes (groundclutter, effects of
anomalous beam propagation, and biological scatterers such as birds and insects, etc.), correcting distorted data, and
quantitative estimation of the final data uncertainty. Detailed descriptions and reviews of currently used methods of QC of
radar data, which identify individual uncertainty components and address their combined effects, are commonly found in
literature (see e.g., Steiner and Smith, 2002; Berenguer et al., 2006; Cho et al., 2006; Gourley et al., 2007; Park et al., 2009;
Zhang et al., 2011; Krajewski et al., 2011; Szturc et al., 2012; Ośródka et al., 2014). Many new dual-polarization-based
quality control algorithms are being implemented in radar signal processors and are part of their own software.

The quantity called quality index (*QI*) plays an increasingly important role in quality control of weather radar data
(e.g., Einfalt et al., 2010; Michelson et al., 2014), as it provides quantitative information on the quality of data, but can also
be used to generate more reliable individual products (Ośródka and Szturc, 2015), to produce composite maps (Fornasiero et
al., 2006; Sandford and Gaussiat, 2012; Jurczyk et al., 2020a) or QPE based on multi-source information (Jatho et al., 2010;
Jurczyk et al., 2020b; Méri et al., 2021).





Huge effort to resolve quality-related issues in radar observations has been made by various international research
programmes, such as COST-717 (Michelson et al., 2005) and COST-731 (Rossa et al., 2010), EUMETNET OPERA
(Huuskonen et al., 2014; Saltikoff et al., 2019), the BALTRAD project (Michelson et al., 2018), and others.
Different QC systems are designed to suit specific local conditions and are implemented in individual national
meteorological services, e.g. ROPO in Finland (Peura, 2002), NMQ in NOAA, USA (Zhang et al., 2012), RADVOL-QC in
Poland (Ośródka 2014), qRad in Slovakia (Méri et al., 2021).
In the Polish national meteorological and hydrological service, i.e. the Institute of Meteorology and Water
Management – National Research Institute, the RADVOL-QC system works operationally to perform QC of radar data
delivered by the Polish weather radar network POLRAD (Ośródka et al., 2014; Szturc et al., 2018). The main objective of
this study is to present new, more effective algorithms incorporated into the RADVOL-QC system which are able to
effectively deal with the abovementioned disturbances in radar data.
The paper is structured as follows: first, the RADVOL-QC system, its structure and applied approaches are briefly
presented (Section 2); then new solutions in the field of non-meteorological echo detection, incorporated into the RADVOL-
QC after the publication describing its earlier version, are discussed in Section 3, which is the essential part of the paper;
next, algorithms for correcting these detected distorted echoes are described (Section 4); finally, there is a brief description
of the verification of the system's effectiveness (Section 5), followed by a concise summary (Section 6).
**2. Description of the RADVOL-QC system for the Polish weather radar network**
**2.1. Polish weather radar network POLRAD**

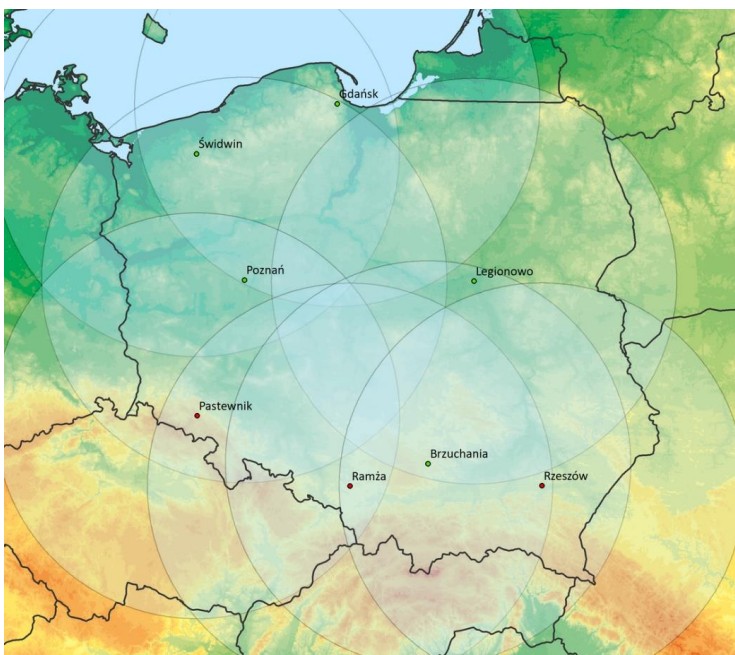

**Figure 1: Coverage of the area of Poland by weather radar network POLRAD with 250-km range as of 2021.**
The algorithms for quality control of weather radar data described in the paper have been designed for Polish weather radar
network POLRAD, operated by the Institute of Meteorology and Water Management – National Research Institute (IMGW).
The network consists of eight C-band Doppler radars (Fig. 1) from which three are polarimetric. All the radars were
manufactured by European company Leonardo S.p.A., formerly Gematronik (Germany). The main parameters of the radars



and designed scan strategy are listed in Table 1. There is a plan in place to replace all radars in this network with dual-
polarization ones and to expand it by two more radars in the years 2022-2023.

**Table 1. Parameters and scan strategy for radar reflectivity scans used in weather radars of the POLRAD network.**

| Parameter | Value |
|---|---|
| Frequency (band) | 5.5 GHz (C-band) |
| Beam polarization | Single (5) and dual (3) |
| Beam width | 1° |
| Sampling along the radar beam | 1 km |
| Number of azimuths | 360 |
| Maximum range | 250 km |
| Number of elevations | 10 |
| Elevation angles | 0.5, 1.4, 2.4, 3.4, 5.3, 7.7, 10.6, 14.1, 18.5, and 23.8° |
| Temporal resolution | 10 min |

**2.2. Structure of the RADVOL-QC system**
The 3-D raw data, so-called volumes, generated by POLRAD radars are quality controlled by the RADVOL-QC system. The
RADVOL-QC quality control system includes data correction and determination of *QI* resulting from each recognized error
source (Ośródka et al., 2014). The QC applies to each radar bin – a point in space that is assigned a single measurement. In
the case of dual-polarization measurements, which offer more possibilities for efficient action but require different
techniques (Bringi et al., 2011), the quality control algorithms constitute a separate category and their names are prefixed
with "DP". Quality control of Doppler data, i.e. velocities, also carried out in RADVOL QC is not described in this paper.
The different kinds of errors taken into consideration by the RADVOL-QC system can be divided into several groups.
*Errors connected with radar beam geometry* and effects related to increasing distance from the radar site along the
beam, such as beam broadening, increasing distance between neighbouring bins and increasing height of the radar beam
above ground level. As a result, extrapolation to the Earth's surface is burdened with higher and higher errors along with
distance from the radar site. Thus, to compensate rainfall underestimation due to these range-related problems, a correction is
performed. It involves two factors: the altitude of the radar beam above ground level and a scaling function that modifies the
correction magnitude depending on the precipitation type, i.e. limits its value for convective precipitation.
*Disturbance by non-meteorological echoes.* Such kinds of echoes are mainly caused by (i) groundclutter (mountains
or high artificial structures located close to the radar site), (ii) external signals from RLAN emitters or the sun interfering
with radar signals; these external signals are usually visible in the radar image as spikes pointing from the emitter towards
the radar site, and can be removed using a geometrical algorithms that analyse the variability of the echo along and across
the radar beam, (iii) small-scale measurement noise, so called speckles, and (iv) biological targets such as birds or insects.
*Other groups of errors* result from (i) beam blockage on terrain (mountains) causing a decrease in radar signal, which
is corrected using a digital terrain map, (ii) attenuation in rain, especially if it is heavy, corrected by iterative estimation of
signal loss along the beam path in cases of single-polarization radars and by an algorithm based on polarimetric information
in cases of dual-polarization radars, and (iii) anomalous propagation of the radar beam which causes, for example, echoes
received from non-meteorological (terrain) and meteorological (hydrometeors) objects located outside the radar range.
Each category of errors burdening radar data is characterized by specific properties, especially spatial and temporal
structure. As a result, different approaches for their diagnosis and correction are required (Table 2).
**Table 2. List of quality control algorithms employed in the RADVOL-QC system for radar reflectivity data measured by single-**
**and dual-polarization (dual-pol) weather radars in order of the data processing chain.**

| Algorithm | Task | Correction technique |
|---|---|---|
| | | |


| BROAD | Quality characterisation due to broadening of the radar beam | Analysis of geometry of the radar beam cross-section |
|---|---|---|
| DP.TURBINE | Detection of non-meteorological echoes, especially from wind farms, for dual-pol radars | Fuzzy logic algorithm based on dual-pol parameters: $Z_{DR}$, $\rho_{HV}$, sd(Z), sd($Z_{DR}$), sd($\Phi_{DP}$) |
| DP.NMET | Detection of non-meteorological echoes: biological, anomalous propagation, etc., for dual-pol radars | Decision tree based on dual-pol parameters: $\rho_{HV}$, sd($\Phi_{DP}$), and sd($Z_{DR}$) |
| TURBINE | Detection of non-meteorological echoes, especially from wind farms and other obstacles | Long-term analysis of radar echoes at the lowest elevation (static masks) |
| SPIKE | Detection of non-meteorological echoes of spike type because of RLAN interference | Analysis of variability of radar echoes across and along the radar beam |
| INTERP | Corrections: interpolation of non-meteorological echoes detected by the algorithms: DP.TURBINE, DP.NMET, TURBINE, and SPIKE | Values from a neighbouring higher elevation after adjustment to the vicinity mean (for DP.TURBINE, DP.NMET, and TURBINE). Interpolation from not burdened neighbouring bins across the radar beam (for SPIKE) |
| NMET | Removal of non-meteorological echoes: biological, caused by anomalous propagation, etc. | Analysis of altitude and intensity of echoes |
| SPECK | Removal of measurement noise (speckles) | Analysis of a given bin vicinity and analysis of a spatial distribution of echo intensity |
| MHV | Compensation of underestimation due to distance to the Earth's surface | Corrections due to extrapolation down to the Earth's surface |
| BLOCK | Correction of the partial and total radar beam blockage by terrain, buildings, etc.; detection of groundclutter | Analysis of the geometry of the radar beam and digital terrain map |
| DP.ATT | Correction of the radar beam attenuation in precipitation for dual-pol radars | Analysis of dual-pol parameters |
| ATT | Correction of the radar beam attenuation in precipitation for single-pol radars (only one of these two: DP.ATT and ATT can be used for a given radar) | Analysis of attenuation along the path of the radar beam (based on specific attenuation) |

Individual quality indices ($QI_i$) are determined in the frame of each $i$-th RADVOL-QC algorithm separately and after the whole quality control chain the final total quality index $QI$ is determined using multiplicative formula (Einfalt et al., 2010; Ośródka et al., 2014). Its values range from 0 for extremely bad to 1 for perfect quality data.

In this paper only the new and enhanced algorithms implemented into the RADVOL-QC system after publication of the earlier paper by Ośródka et al. (2014) are described in detail. Since this time the system has continued to be developed, including within the European BALTRAD project (Michelson et al., 2018) and as part of IMGW's statutory duties.

**3. New solutions for the detection of non-meteorological echoes**

**3.1. New challenges**

The recent dramatic increase in the number of RLAN signals interfering with radar echoes (e.g., Saltikoff, 2016) as well as the greater number of wind farms used for energy generation (e.g., Hood et al., 2010) has created new and serious challenges to the developed QC algorithms, despite the fact that the commonly used approaches show good performance in many observed cases.

The first problem, that of radio interference, is becoming more and more difficult to handle because of a growing number of such echoes. A team of experts from around the world representing institutions involved in weather radar measurements, including the EUMETNET OPERA programme, issued the following drastic statement: "As an extreme solution, and on a theoretical basis, an entire C-band network could be replaced by a dense network of X-band radars (...)" (Saltikoff et al., 2016). Nonetheless, the interference must be removed from radar-based precipitation data before ingestion into any application, such as input into hydrological rainfall-runoff modelling or assimilation into mesoscale numerical weather prediction models. A significant problem with the RLAN-derived echoes, so called spikes, is that although they may be partially removed or decreased due to the low threshold of signal quality index (SQI) of radar data, this in fact complicates their removal because it disrupts their spatio-temporal pattern applied in the relevant algorithm.

As a response to the increasing demand for renewable energy, the number of wind turbines is growing rapidly in many countries around the world. Their impact on weather radar performance has been extensively studied in recent years,



with several cases of wind turbine clutter observed in meteorological radar applications (Isom et al., 2009; Norin and Haase,
2012; Angulo et al., 2015; Norin, 2015). The main objective of these studies was to characterize and try to mitigate wind
turbine clutter, mainly by means of digital signal processing such as clutter-filtering techniques. However, the Doppler
information that is often used for the removal of stationary target returns is not likely to be appropriate in this case because
wind turbine blades are constantly rotating and, consequently, provide a non-zero Doppler signature (e.g., Angulo et al.,

2015).

Erroneously estimated rainfall amounts propagate through hydrological models, affecting hydrologic simulations and
predictions. To mitigate or eliminate the aforementioned RLAN interference and effects of wind turbines in quantitative
rainfall estimation and subsequent applications, this paper proposes the incorporation of enhanced automated approaches
into the RADVOL-QC system.
**3.2. Fuzzy logic scheme for polarimetric parameters – DP.TURBINE algorithm**
The algorithm named DP.TURBINE is used mainly for the detection of radar echoes generated by wind turbines; it employs
a fuzzy logic scheme for selected polarimetric parameters (the definition of dual-polarimetric parameters can be found e.g.
in Bringi and Chandrasekar, 2001). Preliminary analysis has been conducted to check which polarimetric parameters are
most sensitive to non-meteorological echoes. This was investigated by means of histograms for the following classes of radar
echoes: non-meteorological and meteorological, where the latter is divided into two types: convective and stratiform
precipitation. Three parameters were finally selected to be employed in this algorithm: the differential reflectivity factor
$Z_{DR}$(dB), the cross-correlation coefficient between horizontally and vertically polarized radar returns $\rho_{HV}$, and the
differential phase $\Phi_{DP}$(°). The $\rho_{HV}$ is employed in the algorithm directly, whereas for the others standard deviations sd($Z_{DR}$)
and sd($\Phi_{DP}$) computed within grids of 3 bins x 3 bins are employed. Histograms of the selected parameters obtained for
Ramża radar from two days – one with a convective event and the second with a stratiform one – are presented in Fig. 2.

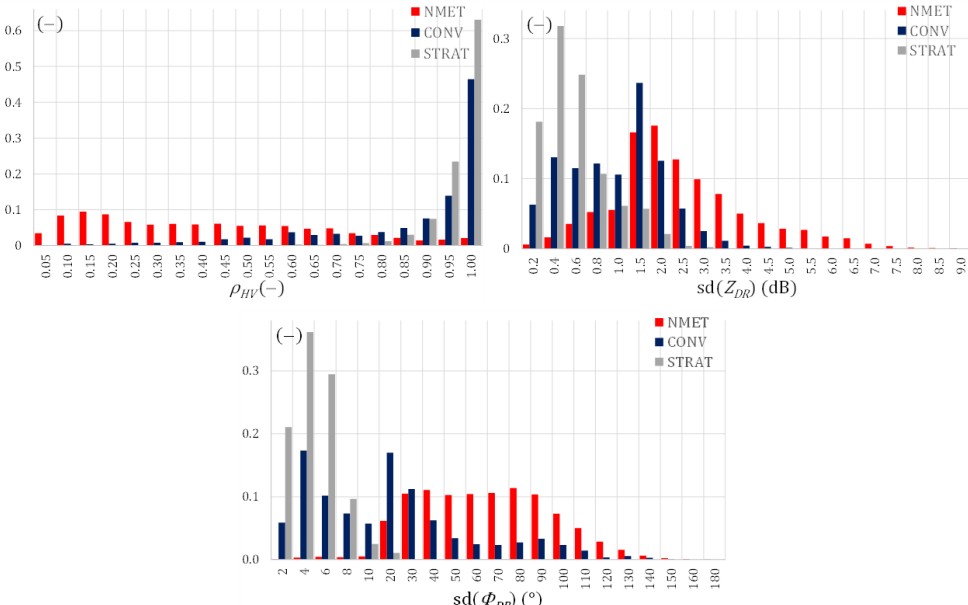

**Figure 2: Histograms for selected polarimetric parameters from the top-left: $\rho_{HV}$, sd($Z_{DR}$), and sd($\Phi_{DP}$), for three classes of radar**
**echoes: non-meteorological (NMET, in red), meteorological convective (CONV, in navy), and meteorological stratiform (STRAT,**
**in grey). Ramża radar, data from two days: 1 September 2018 and 21 February 2019, the lowest elevation.**



It is notable that for correlation coefficient $\rho_{HV}$, the non-meteorological echoes generate values clearly different to
those for meteorological echoes, and the range of overlap is quite narrow. The standard deviations $\mathrm{sd}(Z_{DR})$ and $\mathrm{sd}(\Phi_{DP})$
also differ in values for the three classes, especially for stratiform precipitation. The above information collected for these
three parameters allows one to deduce the presence of non-meteorological echoes with a relatively high certainty.

**Figure 3: Radar measurement fields from the top-left: $Z$, $\rho_{HV}$, $\mathrm{sd}(Z_{DR})$, and $\mathrm{sd}(\Phi_{DP})$. Ramża radar, the lowest elevation, 21 February 2019, 16:40 UTC. At the bottom: definition of axes in polar coordinates.**

In Fig. 3 examples of radar reflectivity $Z$ along with the three polarimetric parameters investigated in the
DP.TURBINE algorithm are presented. All the radar pictures shown in this article, with the exception of those shown in Fig.
12, are presented in the polar coordinates: the horizontal axis gives the azimuth ($\alpha$) in degrees, and the vertical axis gives the
distance from the radar site ($l$) in km.
The evident non-meteorological spike echoes are especially visible on $\rho_{HV}$ data, whereas non-meteorological from
terrain and other obstacles are mainly visible on $\mathrm{sd}(Z_{DR})$ data.
Having selected the most appropriate polarimetric parameters, a fuzzy logic approach was applied to categorize radar
bins into meteorological and non-meteorological echoes. Calibration of the scheme consists in the determination of the
function parameters for each class of echoes: meteorological and non-meteorological, including those from wind farms (Fig.
4).

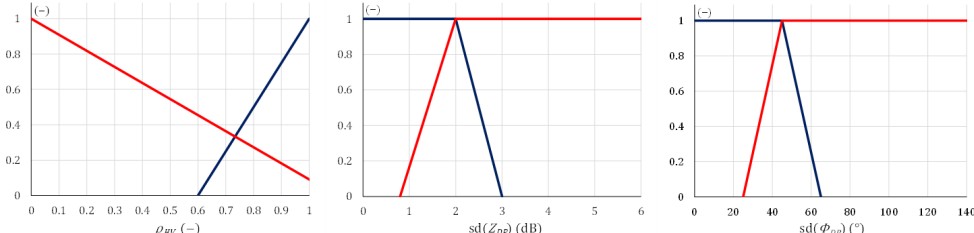

**Figure 4: Membership functions for selected polarimetric parameters from the left: $\rho_{HV}$, sd($Z_{DR}$), and sd($\Phi_{DP}$) determined for**
**meteorological (in navy) and non-meteorological echoes (in red).**
For both echo classes, values of relevant membership functions for all three selected parameters are aggregated as
weighted sums:
$$\mu(class) = \sum_{i=1}^{3} \mu_i(class) \cdot W_i(class) \qquad (1)$$
where $class$ is the echo class (meteorological or non-meteorological), $i$ is the parameter number, 3 is the number of
parameters, $\mu_i(class)$ is the membership function for $i$-th parameter for echo class $class$, and $W_i(class)$ is the weight of $i$-
th parameter for echo class $class$. These weights equal 1.0, 0.5, and 1.0 for parameters $\rho_{HV}$, sd($Z_{DR}$), and sd($\Phi_{DP}$),
respectively. Comparison of the weighted sums decides which echo class a considered radar bin belongs to.
The designed algorithm proved to be effective, especially for the detection of wind turbines. The effectiveness of the
algorithm was checked on data in the form of monthly precipitation accumulation determined for the lowest antenna
elevation (0.5°), because at a higher elevation the number of this kind of non-meteorological echo is not significant. The
radar reflectivity was transformed into precipitation using the Marshall-Palmer formula. The following categories of objects
were identified based on a map of aeronautical obstacles of Poland provided by the Polish Civil Aviation Authority
(http://caa-pl.maps.arcgis.com/apps/webappviewer/index.html?id=252d2be2e6104adcb9be8201660a05b3/): (i) wind
turbines and wind farms, (ii) other disturbing objects, and (iii) undisturbed locations. Effectiveness (*effectiveness*) of the
DP.TURBINE algorithm has been assessed by the formula:
$$effectiveness = 1 - \frac{R_{DP.TURBINE} - R_{bg}}{R_{raw} - R_{bg}} \qquad (2)$$
where $R_{raw}$ is the monthly uncorrected precipitation accumulation at the location of a given object; $R_{DP.TURBINE}$ is the
precipitation accumulation in the same place after using the DP.TURBINE algorithm for the detection of obstacles and then
the INTERP algorithm (see Section 4) for their correction; $R_{bg}$ is the monthly precipitation accumulation determined for the
undisturbed neighbourhood of a considered object.

**Table 3. Monthly precipitation accumulations determined for the lowest elevation (0.5°) at locations of wind turbines, other**
**disturbing objects, and undisturbed locations. Data from Ramża radar, September 2018.**

| Kind of object | Azimuth (°) | Dist. from the radar site (km) | $R_{raw}$ (mm) – raw data | $R_{DP.TURBINE}$ (mm) – after DP.TURBINE | $R_{bg}$ (mm) – background | *effectiveness* of the algorithm |
|---|---|---|---|---|---|---|
| Wind turbines | 16 | 54 | 1371 | 302 | 90 | 0.83 |
| | 331 | 38 | 1060 | 214 | 110 | 0.89 |
| | 270 | 68 | 1096 | 345 | 180 | 0.82 |
| | 253 | 41 | 745 | 189 | 150 | 0.93 |
| | 237 | 41 | 1359 | 264 | 140 | 0.90 |
| | 251 | 56 | 842 | 329 | 210 | 0.81 |
| | 233 | 91 | 784 | 401 | 280 | 0.76 |
| | 229 | 88 | 1302 | 708 | 270 | 0.58 |
| Other disturbing objects | 20 | 56 | 1161 | 272 | 90 | 0.83 |
| | 319 | 43 | 1486 | 279 | 135 | 0.89 |
| | 313 | 51 | 1874 | 166 | 100 | 0.96 |



|  | 246 | 95 | 3543 | 1293 | 220 | 0.68 |
| --- | --- | --- | --- | --- | --- | --- |
| Undisturbed locations | 100 | 79 | 189 | 187 | 187 | 1.00 |
|  | 280 | 79 | 170 | 170 | 170 | 1.00 |
|  | 10 | 119 | 149 | 137 | 137 | 1.00 |
|  | 55 | 39 | 165 | 152 | 152 | 1.00 |


Table 3 shows the influence of the DP.TURBINE algorithm on the values of monthly accumulation of radar echoes in
places where wind turbines and other obstacles (masts, chimneys) disturb the radar observations from Ramża radar. There
are many such objects here and their monthly precipitation accumulations resulting from disturbance from these turbines
often significantly exceed a thousand mm. The *effectiveness* of the developed algorithm for echoes from wind turbines in
range of the Ramża radar, presented in Table 3, is on average 0.82. However, if the intensities of echoes from wind turbines
are lower, this efficiency is lower.
In Fig. 5 an example of the DP.TURBINE running is also presented for the Ramża radar. Within this radar range a lot
of obstacles can be visible in the monthly accumulations, but only a small number of them are due to wind turbines. The
extremely dense strong echoes placed within a distance approximately 30 km from the radar site are a result of the industrial
and urbanized area (the Upper Silesia conurbation) located close to the Ramża radar. The existence of high buildings
produces such strong echoes due to side lobes from the radar beam. Other disturbances are caused by RLAN interference.
The efficiency of the DP.TURBINE algorithm is significantly visible in the right picture. It should be noted that the
SPIKE algorithm for RLAN echo removal was also employed here. Example locations of wind turbines visible in the raw
radar image are marked with a red ellipse.

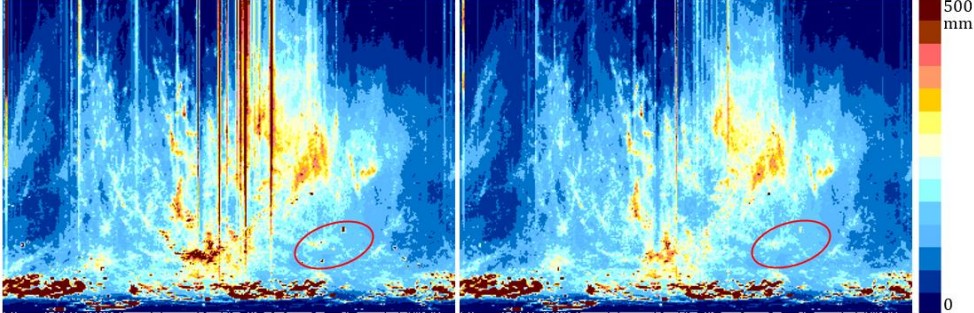


**Figure 5: Example of the performance of the DP.TURBINE algorithm from the left: raw data and data after detection of non-**
**meteorological echoes by means of DP.TURBINE and SPIKE, and their interpolation with INTERP. Ramża radar, monthly**
**accumulation of precipitation, the lowest elevation, September 2018. The red ellipse shows examples of echoes from wind turbines.**
The individual quality index $QI_{DP.TURBINE}$ depends on the presence of a detected non-meteorological echo in a given
bin according to equation:
$$QI_{DP.TURBINE} = \begin{cases} 0.75 & \text{bin with non-meteorological echo} \\ 1 & \text{bin without non-meteorological echo} \end{cases} \qquad (3)$$
**3.3. Decision tree for polarimetric parameters – DP.NMET algorithm**
As opposed to the DP.TURBINE algorithm, which has been designed especially for wind turbine echoes, the DP.NMET is
dedicated to different types of disturbances in weather radar measurements, caused not only by wind turbines but also by
RLAN interference, anomalous propagation, biological objects, etc. The main idea proposed by Seo et al. (2015) is to use a
decision tree, in which the following polarimetric parameters are employed: $Z$, $\rho_{HV}$, $sd(\Phi_{DP})$, and $sd(Z_{DR})$. The adaptation
of this original scheme came down to tailoring parameters to local conditions and excluding the $sd(Z_{DR})$ parameter (Fig. 6).





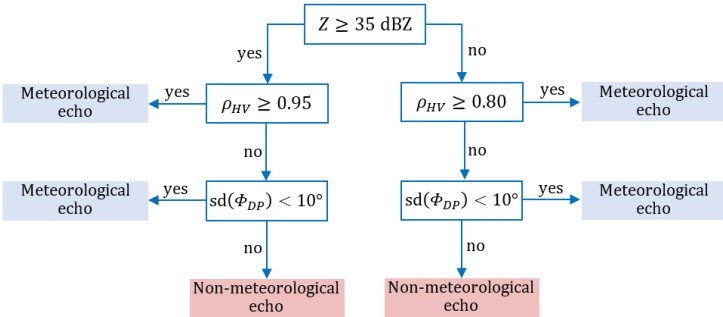


**Figure 6: Flowchart describing the DP.NMET algorithm designed to detect non-meteorological echoes of different types for dual-polarization radars (based on the scheme from Seo et al., 2015).**

This flowchart can be written as one condition:
If $\{[(Z \geq (Z)_{thr})$ and $(\rho_{HV} < (\rho_{HV})_{thr1})]$ or $[(Z < (Z)_{thr})$ and $(\rho_{HV} < (\rho_{HV})_{thr2})]\}$ and $[\mathrm{sd}(\Phi_{DP}) \geq$
$\mathrm{sd}(\Phi_{DP})_{thr}]$ then a given echo is non-meteorological                        (4)
where values of the thresholds have been estimated for the radars of the POLRAD network as follows: $(Z)_{thr} = 35$ dBZ,
$(\rho_{HV})_{thr1} = 0.95$, $(\rho_{HV})_{thr2} = 0.80$, and $\mathrm{sd}(\Phi_{DP})_{thr} = 10°$.
The main approach of the algorithm is that the basic polarimetric parameter for differentiation between
meteorological and non-meteorological echoes is the correlation coefficient $\rho_{HV}$: the echo is non-meteorological if the value
is lower than a defined threshold, however for lower reflectivity the threshold is decreased, i.e. fewer echoes are classified as
non-meteorological. Because the $\rho_{HV}$ has a wide overlapping range for the meteorological and non-meteorological classes
(see Sect. 3.2), additional verification is needed since false detection of non-meteorological echoes occurs too often. For this
reason, in the next step the standard deviation $\mathrm{sd}(\Phi_{DP})$ is used to confirm the preliminary classification: if the value is higher
than a defined threshold, the echo is recognized as non-meteorological.

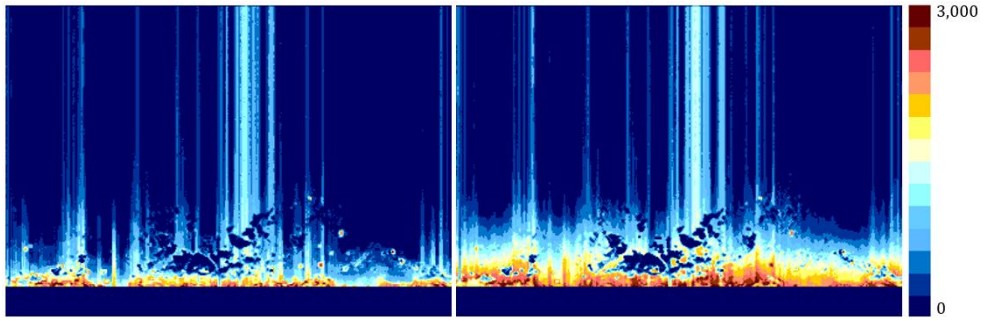


**Figure 7: Numbers of non-meteorological echoes detected separately by algorithms from the left: DP.TURBINE and DP.NMET. Ramża radar, September 2018, the lowest elevation. Up to 25 km from the radar, the algorithm is not applied.**

Fig. 7 shows how many times the DP.TURBINE (discussed in Sect. 3.2) and DP.NMET algorithms separately
detected non-meteorological echoes during one month for Ramża radar data at the lowest elevation. The total number was 3
thousand out of approximately 4,320 measurements.
Both algorithms are not applied up to 25 km from the radar site because within the RADVOL-QC system one of the
SPECK sub-algorithms (see Ośródka et al., 2014) operates in this area, replacing the data on the lowest elevation with data
from the adjacent higher one, which explains the wide strip near the radar site where no echoes were detected.
It turned out that the DP.TURBINE algorithm is better at extracting small-area echoes associated with wind turbines,
while the DP.NMET algorithm detects more other non-meteorological echoes, especially those located near the radar (but





outside the aforementioned 25-km strip) caused by side lobes of the radar beam hitting various types of obstacles. It is
noticeable that both algorithms are also sensitive to other types of non-meteorological echoes, especially those connected to
RLAN interference of spike type. As mentioned in Section 3.2, the Ramża radar generates the most disturbed data, especially
in the vicinity of the radar, due to its location near a large industrial and urban area.

Similarly to the previous algorithm, the individual quality index $QI_{DP.NMET}$ depends on the presence of a detected

non-meteorological echo in a given bin:
$$QI_{DP.NMET} = \begin{cases} 0.75 & \text{bin with non-meteorological echo} \\ 1 & \text{bin without non-meteorological echo} \end{cases} \qquad (5)$$
**3.4. Static maps (masks) of echoes – TURBINE algorithm**
The algorithm using static maps (masks) of groundclutter of various origins is a very simple but effective tool. However, it
requires relatively frequent updating of maps depending on the intensity of obstacle changes in the range of a given radar.
Fig. 8 shows an example of such a mask for the Legionowo radar located in central Poland. It is near Warszawa (Warsaw)
city, which causes a significant number of obstacles to the south of the radar site. Moreover, echoes from various kinds of
wind turbines are visible here.

The following algorithm has been developed for the semi-automatic generation of static echo maps from wind

turbines and other groundclutter. At the first stage, fields of precipitation are generated for non-rainy time-steps (when a
number of rainy bins is below a defined threshold) for the lowest elevations of each radar. Every few months they are
accumulated, and on this basis maps of permanent echoes from wind farms as well as residential and industrial buildings are
generated for bins exceeding a certain threshold value, specific for each radar. At the last stage, the maps are manually
corrected by comparing them with the initial fields and with previous masks in order to exclude from the masks echoes such
as those from the mountains to avoid significant reduction of precipitation in these areas.

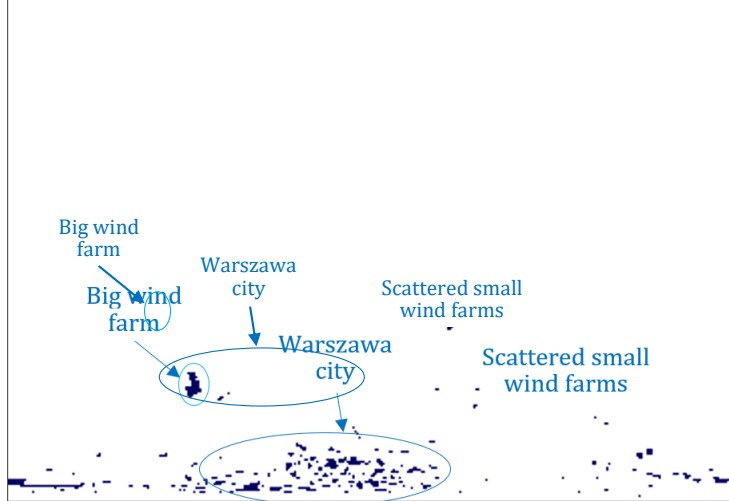

**Figure 8: Example of a mask of non-meteorological permanent echoes. Legionowo radar, the lowest elevation.**

Figure 9 shows the monthly precipitation sums for selected radars of the POLRAD network: from raw data and after

processing with the RADVOL-QC system without and with the TURBINE algorithm; the latter stands for the complete
system. The RADVOL-QC system proved to be very effective even without the use of the TURBINE algorithm because
some other algorithms, especially those based on data from dual polarization radars, detect echoes from wind turbines as
well. This improvement is especially noticeable for radars covering mountainous areas, such as Pastewnik and Ramża, as





well as highly urbanized areas such as Legionowo and Ramża. More effective removal of wind farms employing the
TURBINE algorithm for Legionowo can be explained by the fact that this radar is not dual-polarimetric.

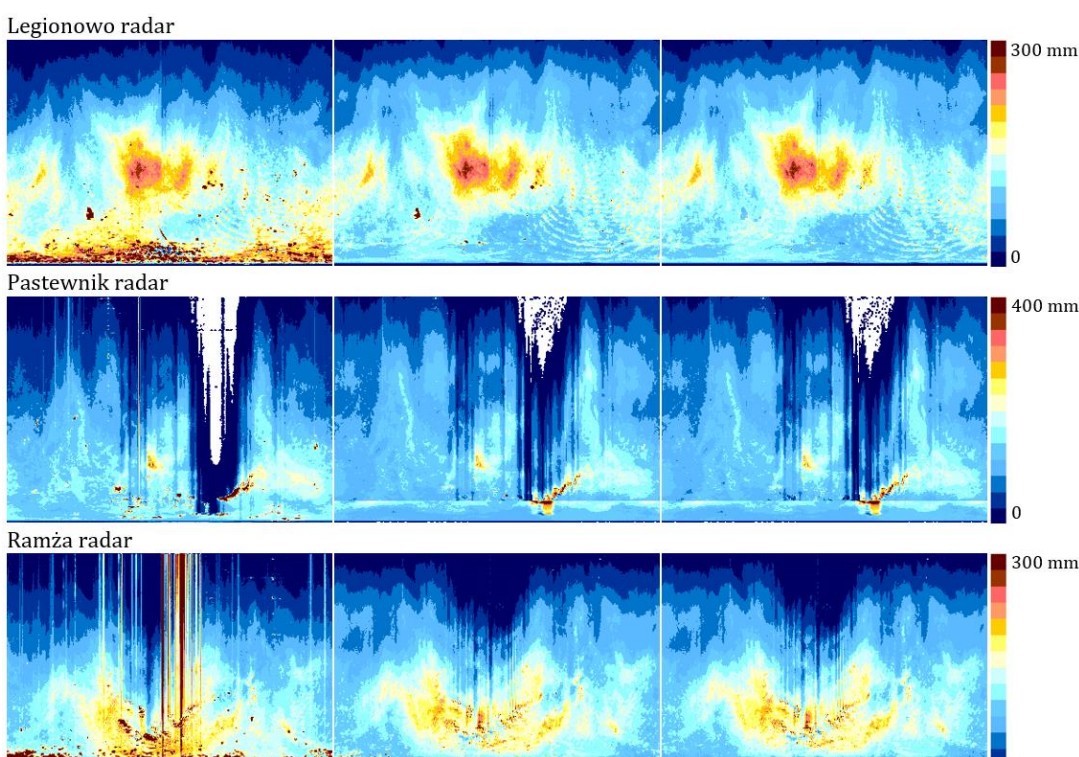

**Figure 9: Comparison of the monthly sums of radar precipitation from (from the left): raw data, data after processing by the**
**RADVOL-QC system without the TURBINE algorithm, and with the TURBINE algorithm (i.e. the complete RADVOL-QC**
**system). Legionowo, Pastewnik, and Ramża radars, the lowest elevations. October 2019.**

The individual quality index $QI_{TURBINE}$ is lowered in the bins located within the mask of permanent non-
meteorological echoes:
$$QI_{TURBINE} = \begin{cases} 0.75 & \text{bin with non-meteorological echo} \\ 1 & \text{bin without non-meteorological echo} \end{cases} \qquad (6)$$

**3.5. Geometrical algorithms for the removal of RLAN interference – SPIKE algorithm**
The Polish radar network POLRAD consists of eight radars, but only three of them are equipped with the functionality of
dual-polarisation of the radar beam. Therefore, algorithms that do not use polarimetric parameters to remove non-
meteorological echoes are still very important to ensure the appropriate quality of data, especially as not all measurement
disturbances can be removed through applying polarimetric data.
In C-band weather radars, signals generated by external RLAN systems, and also from the sun – so-called spike
echoes – are interpreted by radar as precipitation echoes. The shape of these echoes is very specific: in Cartesian radar
images they have the form of elongated narrow spike-shaped echoes, located along the radar beam, sometimes with high
reflectivity.
The SPIKE algorithm actually consists of a set of sub-algorithms that are sensitive to the various properties of this
type of radar echo. Since this algorithm has been an important element of the RADVOL-QC system from the very





beginning, it has been described in detail in the earlier work of Ośródka et al. (2014). However, it has undergone major
changes since then, so in this paper the areas where the most significant changes took place will be described in greater
detail.

In the algorithm for eliminating this type of echo from radar pictures, the spatial structure of the reflectivity field is
analysed separately for each elevation. The algorithm is divided into several sub-algorithms used to remove different types
of spike echoes, which are named in simple terms:
–    "wide",
–    "narrow",
–    "discontinuous",
–    "shorter longitudinal",
–    "inverse".

*Sub-algorithm for "wide" spike detection*
This sub-algorithm is used for spikes if the echo percentage coverage of a given elevation does not exceed a pre-set threshold
value. The echo changeability is examined by means of its local variance across and along the radar beam: echoes are
classified as "potential" spikes when the variance of reflectivity across the beam is high and low along the beam. This
algorithm was described in an earlier paper by Ośródka et al. (2014).

*Sub-algorithm for "narrow" spike detection*
The next algorithm is used to detect spike echoes of small width, i.e. not wider than 5º. In the first step, for a given radar bin
in which a radar echo is observed, the reflectivity values in the bins located at the angular distance from a given azimuth and
at the same distance from the radar site are examined if they are sufficiently weak – in this case a potential spike echo is
recognized in this bin. This check is repeated for all possible shorter angular distances as well. This algorithm was also
described by Ośródka et al. (2014).

*Sub-algorithm for "discontinuous" spike detection*
This newly introduced sub-algorithm is applied to spike echoes of generally elongated shape, but discontinuous along and
across the radar beam. The sub-algorithms described above cannot manage with such spikes due to discontinuities in their
pattern.

The algorithm consists of two steps: at first potential azimuths with "discontinuous" spikes are detected, then they are
confirmed by appropriate changeability of the radar beam. For the lowest elevation the first step is omitted because this scan
can be very rainy, so detection of potential spikes may not be efficient and the azimuths with potential spikes are taken from
the neighbouring higher elevation.

In the below procedure for each azimuth $\alpha$ for a given elevation $\varepsilon$ (apart from the lowest one) the echo bins are
counted along the whole radar beam to obtain the $n(\varepsilon, \alpha)$ values:
$$n(\varepsilon, \alpha) = \sum_{l=1}^{N} i(\varepsilon, \alpha, l), \quad \text{where:} \quad i(\varepsilon, \alpha, l) = \begin{cases} 1 & Z(\varepsilon, \alpha, l) = \text{echo} \\ 0 & Z(\varepsilon, \alpha, l) = \text{noecho} \end{cases} \qquad (7)$$
where $N$ is the number of samplings along the radar beam.

The $p(\varepsilon, \alpha)$ value is determined as the larger of the modulus values of differences between $n(\varepsilon, \alpha)$ and the values in
the adjacent azimuths $\alpha - 1$ and $\alpha + 1$:
$$p(\varepsilon, \alpha) = \max(|n(\varepsilon, \alpha - 1) - n(\varepsilon, \alpha)|, |n(\varepsilon, \alpha + 1) - n(\varepsilon, \alpha)|) \qquad (8)$$

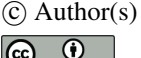



Then for the analysed beam $\alpha$ the two means are calculated: $\overline{p_1(\varepsilon, \alpha)}$ and $\overline{p_2(\varepsilon, \alpha)}$ within the ranges $\pm p_1{}^{thr}$ and
$\pm p_2{}^{thr}$ respectively, and the value $P(\varepsilon, \alpha)$ is determined:
$P(\varepsilon, \alpha) = \overline{p_1(\varepsilon, \alpha)} - \overline{p_2(\varepsilon, \alpha)}$                                               (9)
Potential "discontinuous" spikes are found in the radar beams on azimuths that meet the following condition:
If $(P(\varepsilon, \alpha) > P_{thr1})$ or $[(P(\varepsilon, \alpha) > P_{thr2})$ and $(p(\varepsilon, \alpha) > P_{thr3})]$ then there is a potential spike in the azimuth $\alpha$     (10)
where $P_{thr1}, P_{thr2}$, and $P_{thr3}$ are the threshold values.
Due to the specific shape of this type of echo, an additional check is introduced in order to smooth the field of
relevant potential spikes: azimuth $\alpha$ is considered a potential spike if azimuths $\alpha - 1$ and $\alpha + 1$ are potential ones.
In the final step, bins in azimuths with such potential spike echoes are confirmed for each elevation employing
variances calculated across the radar beam $var_{across}(\varepsilon, \alpha, l)$, also taking into account the "no echo" values:
If $[(var1_{across}(\varepsilon, \alpha, l) > var_{thr1})$ and $(var2_{across}(\varepsilon, \alpha, l) <$
$var_{thr2})]$ then there is a potential spike in the radar bin $(\varepsilon, \alpha, l)$     (11)
where the variance $var1_{across}(\varepsilon, \alpha, l)$ is calculated for reflectivity in dBZ and $var2_{across}(\varepsilon, \alpha, l)$ for reflectivity in
$mm^6 \ m^{-3}$, and $var_{thr1}$ and $var_{thr2}$ are the threshold values.
The concept of the above formula is that "discontinuous" spike echoes have a specific variance of radar reflectivity in
echoes across the radar beam. The variance calculated in dBZ, hence logarithmic $\left(1 \ dBZ = 10 \cdot \log_{10}(1 \ mm^6 \ m^{-3})\right)$, is high
for echoes of this type (Fig. 10, on the left). However, high variance in dBZ is also characteristic of intense meteorological
echoes, especially those originating from convective rainfall. On the other hand, in the case of non-logarithmic values ($mm^6$
$m^{-3}$), the variance of meteorological echoes is relatively high, because their values are generally higher and at the same time
more internally differentiated than that of spike echoes. Thus, the algorithm for the detection of "discontinuous" spike
echoes assumes that they are both characterized by a high variance calculated for reflectivity in dBZ and a relatively low
variance calculated for reflectivity in $mm^6 \ m^{-3}$.
The example in Fig. 10, shows that the criterion associated with high dBZ values clearly indicates radial echoes
located south and slightly west as "discontinuous" spikes, but also at the edges of clearly meteorological echoes. On the
other hand, the criterion based on the values of $mm^6 \ m^{-3}$ prevents meteorological echoes from classification as potential
spike echoes, which could occur in the case of a few small echoes marked by a green ellipse. The red ellipse shows example
of spike echoes. The echoes at the edges of meteorological echoes, which were classified as "potentially" non-
meteorological by both conditions, were finally not confirmed as spikes (see paragraph "Verification of potential spike-type
echoes").

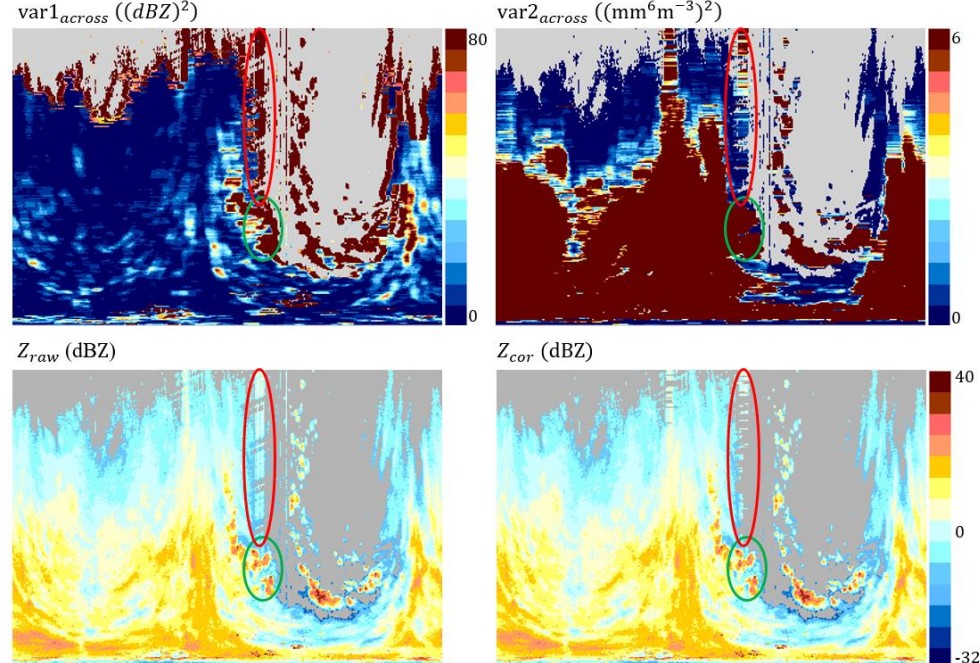


**Figure 10: Example of the performance of the SPIKE and INTERP algorithms. At the top: variances of radar reflectivity across the radar beam calculated from the left, in dBZ and mm6 m-3. At the bottom: radar reflectivity from the left, raw dBZ ($Z_{raw}$) and corrected (interpolated) ($Z_{cor}$). In the red ellipse: "discontinuous" spike echoes; in the green ellipse: meteorological echoes correctly not qualified as spikes. Świdwin radar, the lowest elevation, 9 May 2019, 11:00 UTC.**

*Sub-algorithm for "shorter longitudinal" spike detection*

This also newly introduced sub-algorithm is relatively aggressive, so it is only used for the radars which are extremely burdened with spike echoes, just for the lowest elevation.

For the lowest elevation, using the appropriate threshold for reflectivity $(Z)_{thr}$, for each radar bin $(\varepsilon, \alpha, l)$ the numbers of successively adjacent bins of reflectivity above the threshold, along $size_{along}(\varepsilon, \alpha, l)$ and across $size_{across}(\varepsilon, \alpha, l)$ the given radar beam, are counted. The "shorter longitudinal" spike echo in a bin $(\varepsilon, \alpha, l)$ is detected if the following condition is met:

If $\left( size_{along}(\varepsilon, \alpha, l) > \left( size_{along} \right)_{thr} \right)$ and $(size_{across}(\varepsilon, \alpha, l) <$

$(size_{across})_{thr})$ then there is a "shorter longitudinal" spike in the radar bin $(\varepsilon, \alpha, l)$     (12)

where $\left( size_{along} \right)_{thr}$ and $(size_{across})_{thr}$ are the threshold values.

*Sub-algorithm for "inverse" spike detection*

The term "inverse" spike-type echo means an area in this case of decreased reflectivity values in the shape of a spike aimed at the radar site visible on a radar picture. Such echoes are most often the result of beam blocking, but the BLOCK algorithm does not detect them by means of a digital terrain map.

This sub-algorithm is used only for the two lowest elevations. For each azimuth the radar bins with potential "inverse" spike echoes are found using the following condition:

If $\left( (Z(\varepsilon, \alpha - d, l) - Z(\varepsilon, \alpha, l)) > Z_{thr} \right)$ and $\left( (Z(\varepsilon, \alpha + d, l) - Z(\varepsilon, \alpha, l)) > \right.$

$Z_{thr})$ then there is a potential spike in the radar bin     (13)

Discussion started: 11 October 2021



where $Z_{thr}$ is the threshold value. This procedure is performed for values of $d$ from 1 to $d_{thr}$.

*Verification of potential spike-type echoes*
This SPIKE sub-algorithm plays a verifying role for all the sub-algorithms described above: it is used to check all echoes
that have been flagged as "potential" spike echoes, that is, suspected to be non-meteorological echoes.
For this aim, in a given azimuth $\alpha$ and elevation $\varepsilon$ the number of bins where "potential" spike echoes have been
detected is counted separately for each spike echo type. If a pre-set threshold value has been exceeded, the bins with
"potential" spikes of all types at a given azimuth $\alpha$ and elevation $\varepsilon$ are considered confirmed spikes.
An example of detecting spike echoes, including the "wide", "narrow", and "inverse", is shown in Fig. 11. At the
locations marked in red (in the bottom figure) where such echoes have been detected, interpolation of the values was
performed (see Sect. 5) – the result is shown in the picture on the right.

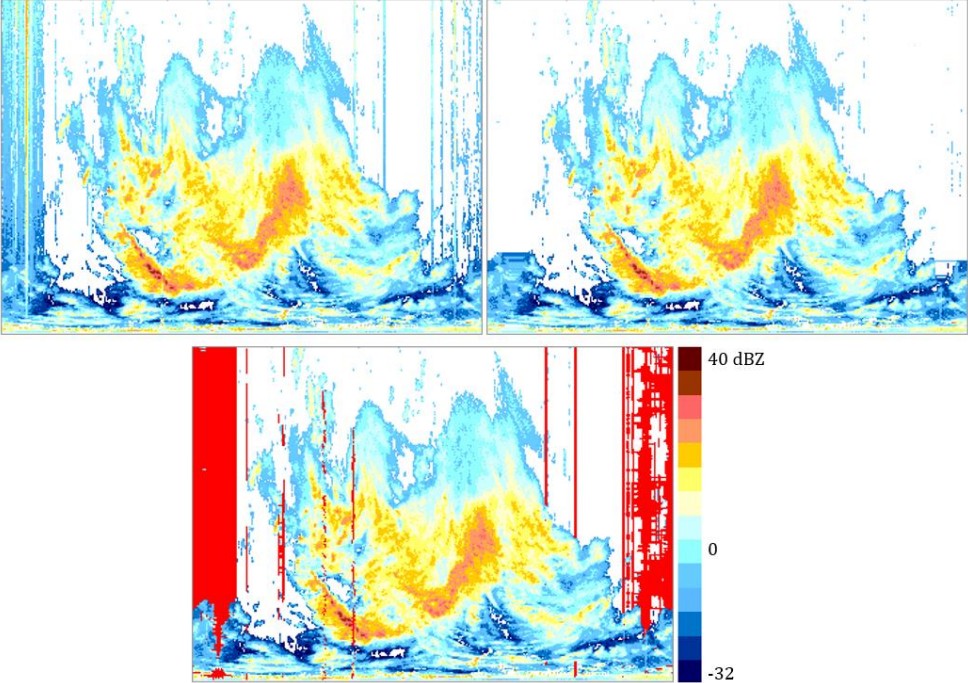


**Figure 11: An example of detecting spike echoes with the SPIKE algorithm using sub-algorithms for "wide", "narrow", and**
**"inverse" spikes, from the left: raw and corrected data, and at the bottom mask (in red) of detected spike echoes. Gdańsk radar,**
**the lowest elevation, 30 September 2019, 14:00 UTC.**
*Quality index*
The $QI_{SPIKE}$ quality index for individual radar bins, and the whole radar beam in which any type of spike has been detected,
is determined from the following formula, in which after fulfilling the first condition in order in the formula, the *QI* value is
decreased and the procedure is stopped:



$$QI_{SPIKE}(\varepsilon, \alpha, l) = \begin{cases} 0.2 & \text{the "wide" spike is detected in a bin } (\varepsilon, \alpha, l) \\ 0.7 & \text{the "wide" spike is detected in a beam } l \\ 0.5 & \text{the "narrow" spike is detected in a bin } (\varepsilon, \alpha, l) \\ 0.8 & \text{the "narrow" spike is detected in a beam } l \\ 0.8 & \text{the "shorter longitudinal" spike is detected in a bin } (\varepsilon, \alpha, l) \\ 0.9 & \text{the "shorter longitudinal" spike is detected in a beam } l \\ 0.5 & \text{the "discontinuous" spike is detected in a bin } (\varepsilon, \alpha, l) \\ 0.8 & \text{the "inverse" spike is detected in a beam } l \\ 1.0 & \text{no spike in a beam } l \end{cases}$$ (14)

### 4. Correction of detected non-meteorological echoes – INTERP algorithm


Non-meteorological echoes detected with the use of specialized algorithms must be replaced with more reliable values of
radar reflectivity. In the RADVOL-QC system this is performed by the INTERP algorithm, which works on radar bins that
have been flagged by the algorithms DP.TURBINE, DP.NMET, TURBINE, and SPIKE as being burdened with non-
meteorological echoes.

### 4.1. Removal of RLAN interference detected by the SPIKE algorithm


Values of radar reflectivity flagged as spikes in bins are interpolates in the following way: for a given flagged bin, the bins in
adjacent and then the further azimuths (on the left and right) at the same distance to the radar site are checked until un-
flagged ones are found. Reflectivity value in the analysed bin is determined as an average from both values found on the left
and right, but if one of the interpolating values is "no echo" then interpolated value is taken as "no echo" as well.

### 4.2. Removal of non-meteorological echoes detected by DP.TURBINE, DP.NMET, and TURBINE algorithms


Values in radar bins in which non-meteorological echoes have been detected by algorithm DP.TURBINE, DP.NMET, or
TURBINE are spatially interpolated. At the top elevation, echoes in the bins flagged as non-meteorological are removed. At
other elevations, flagged adjacent bins are clustered into objects which are divided into ordinary and large ones using a pre-
set threshold.
For ordinary objects the interpolation is performed in the following way: for a particular object at elevation $\varepsilon$ two
values are determined – $Z_{out}(\varepsilon)$, the averaged reflectivity in bins closely surrounding the object in a given elevation ($\varepsilon$) and
$Z_{in}(\varepsilon + 1)$, averaged reflectivity from bins corresponding to the position of the object on the neighbouring higher elevation
($\varepsilon + 1$). The difference $c$ between the two values:
$$c = Z_{in}(\varepsilon + 1) - Z_{out}(\varepsilon)$$ (15)
is employed to retrieve corrected data by subtracting this difference from the disturbed values:
$$Z_{cor}(\varepsilon, a, l) = \min\{Z(\varepsilon + 1, a, l) - c, \ Z(\varepsilon, a, l)\}$$ (16)
For large objects, the difference $c$ is assumed to be zero, and then Equation 16 takes the simpler form:
$$Z_{cor}(\varepsilon, a, l) = \min\{Z(\varepsilon + 1, a, l), \ Z(\varepsilon, a, l)\}$$ (17)

### 5. Verification


The verification of the solutions proposed in Section 3 is presented along with the descriptions of individual algorithms,
because the effectiveness of each algorithm should be assessed separately. Moreover, in this Section the verification of the
performance of the complete RADVOL-QC system is presented.



The effect of disturbances in weather radar data on the uncertainty of the results of the various meteorological and

hydrological models ingesting the data, e.g., in assimilation to mesoscale numerical weather prediction models or as an input

to hydrological rainfall-runoff models (Sokol et al., 2021), depends on the specific application. It is difficult to carry out one

general verification of the effectiveness of algorithms used to quality control this data.

The simplest form of verification is a visual investigation of the effect of corrections on particular time-steps. Fig. 12

shows an example of the combined performance of all the algorithms of the RADVOL-QC system. This example is

presented in the Cartesian system with the radar site in the centre, as the data from a single radar are usually distributed to

applications and end-users in this form. Surface rainfall intensity (SRI) radar product at the height of 1 km above ground

level is employed. In this figure for Ramża radar, on the left echoes from the following disturbances are visible: several

echoes of various types of spikes mainly to the south of the radar site, echoes from the side lobes near the radar, as well as

single non-meteorological echoes observed especially in the eastern part (mainly caused by residential and industrial

buildings, hills and wind farms). In the right picture, the complete RADVOL-QC system copes with correcting various

disturbances in radar data quite effectively.

In the *QI* field for the same time-step, apart from a reduction in the quality of the spike echoes, there is an evident

reduction in the *QI* values in the south – a result of blocking by the mountains at a distance of about 50 km. In addition,

worse quality of measurements is clearly visible with increasing distance from the radar site.


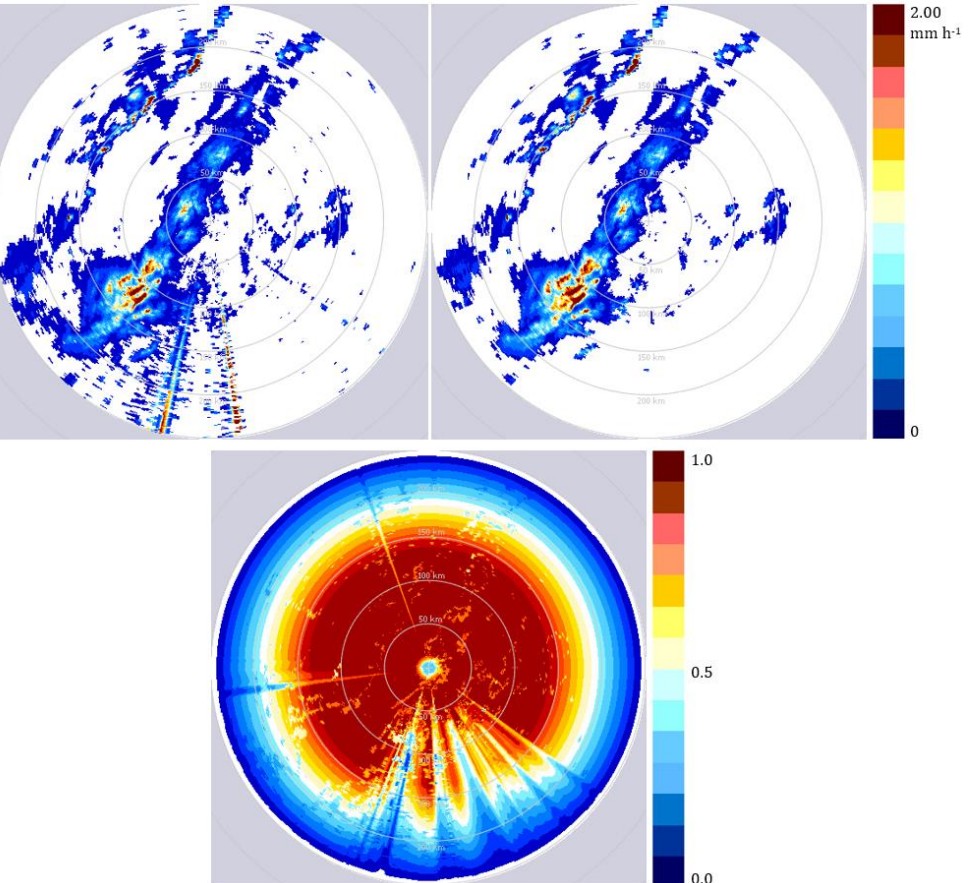

**Figure 12: Example of RADVOL-QC algorithm performance, from the left: raw data, data after corrections made with all**
**algorithms, and related quality index field. Ramża radar, SRI product, 2 October 2019, 12:10 UTC, range 250 km.**





The verification can be performed from the perspective of systems issuing warnings about heavy precipitation. The
graphs in Fig. 13 show the number of exceedances of the threshold value of 1 mm / 10 min on the lowest elevation before
and after RADVOL-QC corrections over one month; areas with more than 200 alarms are marked in red. It is evident the
corrections prevent the generation of false warnings to a large extend.

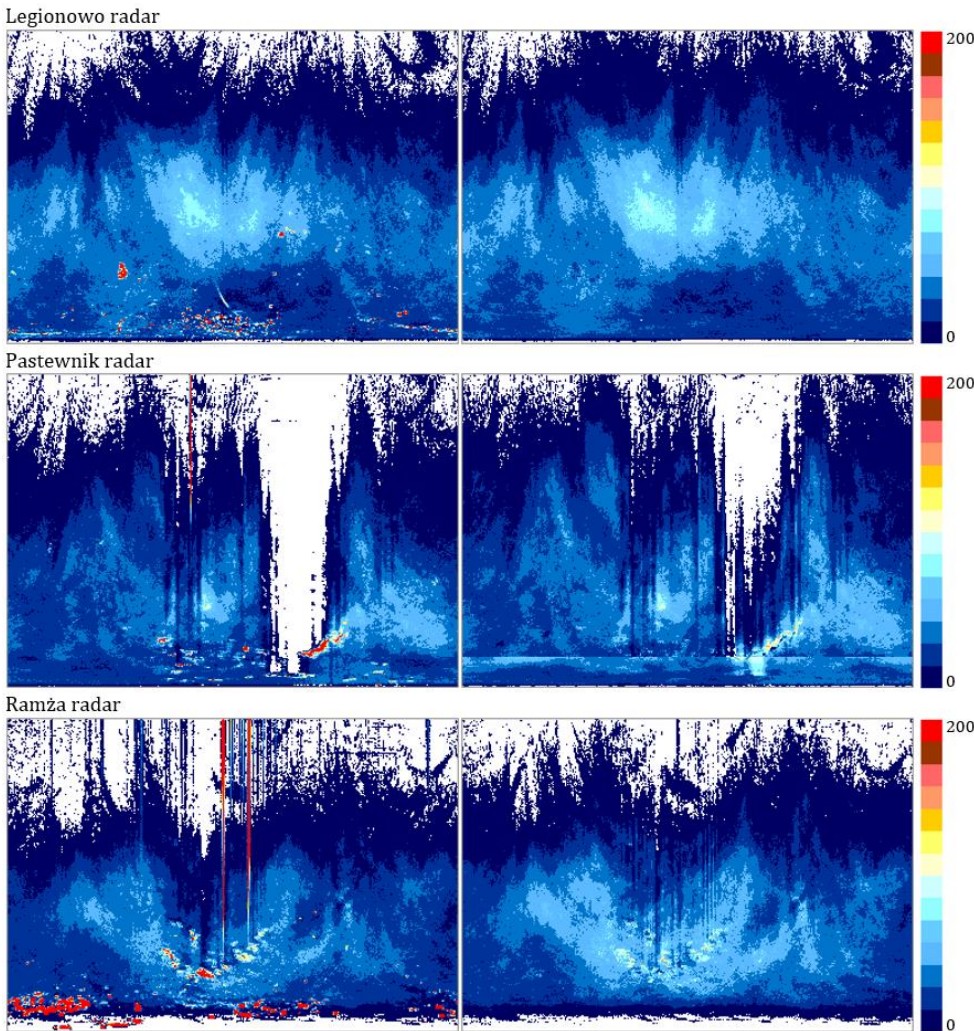

**Figure 13: The graphs of the number of exceedances of precipitation 1 mm / 10 min, from the left: raw data and data after**
**corrections (red denotes over 200 exceedances). POLRAD radars: Legionowo, Pastewnik, and Ramża, the lowest elevations,**
**October 2019.**
All places marked in red can be associated with permanent non-precipitation echoes. These echoes are mainly from
mountain areas and large urban centres. For some radars, these are also echoes caused by signals from RLAN antennas
(Pastewnik and Ramża radars). The relatively extensive echo visible for the Legionowo radar to the east comes from a large
wind farm complex: this is the largest echo from wind turbines visible on the radars of the POLRAD network (see Fig. 8).





## 6. Conclusions

Over recent years a significant increase in the number of external disturbances in radar measurements has been observed, especially those related to RLAN signals interfering with C-band radar signals, as well as to echoes from turbines and wind farms, because their moving parts affect the radar beam in a specific way. On the other hand, a huge advancement in the technology of radar signal processors used in modern weather radars has been observed – these are much better at filtering out non-meteorological echoes than earlier systems, although the problem is still far from being effectively solved.

For these reasons, national meteorological services and various research centres are constantly developing more and more effective algorithms for the detection and removal of non-metrological echoes from radar observations. Software solutions consisting in the analysis of raw 3-D radar data are still an indispensable element of radar data processing systems. Their important feature is that they must be adapted not only to individual types of radars, but also to the local conditions in which they are installed. Algorithms that apply an approach to the problem of eliminating non-meteorological echoes involving the detection and correction of each type of disturbance separately must be constantly developed to take into account their new manifestations.

A large part of this study is devoted to such new challenges. This paper does not describe the entire RADVOL-QC system used to quality control data from the Polish weather radar network POLRAD, as it was already published in detail (Ośródka et al., 2014), but only deals with recently introduced new algorithms responding to the increasing importance of various types of disturbances, in particular those resulting from the growing influence of signals from RLAN installations, as well as the impact of wind turbines grouped into large farms.

There is no doubt that with the emergence of new sources of disturbance and the intensification of existing ones, sustained work on the development of systems such as RADVOL-QC must be continued in parallel with advances in the technology of radar signal processors.

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
