# Peer review of "Improvement in algorithms for quality control of weather radar data (RADVOL-QC system)"

_Atmospheric Measurement Techniques, 2021_

## Author Comment (AC1)

**RC1**: 'Comment on amt-2021-324', Anonymous Referee #1, 01 Nov 2021

> Thank you to the Reviewer for valuable comments! We tried to include all of them in the revised version of our article. Detailed answers to the comments are given below.

The manuscript "Improvement in algorithms for quality control of weather radar data (RADVOL-QC system)" written by K. Ośródka and J. Szturc describes new version of a quality control algorithm designed for radar data of the Polish weather radar network POLRAD. The manuscript in details describes improvements of the current version of RADVOL-QC system in comparison with previous version published in 2014.

The manuscript describes in more detail new algorithms that focus on solving problems mainly due to the influence of wind turbines and RLAN signals, which increasingly negatively affect radar data. In addition to the description of individual algorithms, the manuscript also verifies the efficiency of the described algorithms. This verification is primarily visual. The article would benefit if the verification were performed in some "objective" way, but I am aware that it is not easy and I do not insist on this verification.

> There is a problem with numerical verification because we have no reference, so some indirect methods have to be used. In the previous article from 2014, we introduced two coefficients: "symmetry" and "smoothness", for which sums of precipitation must be determined separately for each radar, long enough to be independent of the influence of local convective precipitation, at least monthly. We then argue that the more homogeneous the precipitation field, the better the radar data is corrected. However, it should be remembered that the impact of few types of disturbances can be diagnosed in this way - only those associated with a permanent location of obstacles (groundclutter, wind farms, etc.) excluding disturbances distributed randomly. So we decided not to present this kind of verification. In our opinion, more is the assessment of the effectiveness of correction algorithms on instantaneous measurements, but due to the aforementioned lack of reference (benchmark), we do not see the possibility of a really objective verification.

The algorithms are described but their parameter values are shown only in one case. It would be nice for readers and potential users of described algorithms to know specific values of thresholds used by the authors.

> We added all values of parameters.

The article is well structured and is written clearly. It is obvious and well known that the authors have been interested in the given topic for a long time and they are experts in this field. I appreciate that the article is based on and solves real operational problems associated with the preparation of radar data. The topic is suitable for the selected journal and will be interesting for people using meteorological radars as they propose new tested algorithms.

I recommend improving the language of the text.

> We tried to improve the text linguistically.

I recommend accepting the manuscript after miner revision.

**Specific comments:**

Line 71 – Please, reformulate "is assigned a single measurement".

> Corrected to: "where a single measurement is performed".

Line 74 – "," should be after RADVOL QC.

Corrected.

Line 78 – I recommend explaining why data should be extrapolated to the Earth.

We have removed the three sentences because they are related to quantitative precipitation estimation which is not discussed in the paper.

Line 117-118 – Could you describe in more detail "this in fact …algorithm".

We agree that the sentence is not clear. We decided to move it to Section dedicated to our corrections of RLAN echoes and to extend it.

Line 141 – I do not consider "event" as an appropriate word in this context. Convective weather?

Corrected.

Line 204 – "significantly" is not suitable word in this context.

Corrected.

Line 232 – Could you clarify your statement about the threshold value?

The sentence is modified.

Line 299-301 – Do you say that after this sentence the modified algorithms are described? It is not obvious to me.

The sentence is modified.

Line 334 – When you speak about echo here and also in other parts of the text do you mean non zero reflectivity or reflectivity exceeding some threshold?

We have added this explanation to the paper on line 246 of the original text.

Line 399 – Please reformulate the sentence "If a …".

The sentence is reformulated.

Line 410-412 – I am not sure that I understand the meaning of the sentence. What is the first condition and when is the procedure stopped?

The sentence is reformulated.

Line 420 – interpolated instead of interpolates.

Corrected.

---

## Author Comment (AC2)

**RC2**: 'Comment on amt-2021-324', Anonymous Referee #2, 19 Nov 2021

> We would like to thank the Reviewer for the careful study of our paper and insightful comments - taking them into account will improve the quality of our paper.

The article "Improvement in algorithms for quality control of weather radar data (RADVOL-QC system)" by Katarzyna Ośródka and Jan Szturc presents some newly developed and enhanced algorithms and methods of filtering and quality evaluation of weather radar data in the RAVOL-QC software developed in the Polish Institute of Meteorology and Water Management, mainly focusing on non-meteorological echo identification. It describes two method using dual-polarization data and two method using single-polarization data. The manuscript addresses actual problems, especially the detection of wind turbines and RLAN interferences. It is well written, clear and the cited literature is relevant and actual. The paper can be considered as very useful for the weather radar scientific community to further enhance the usage of weather radar data. It fits very well the topic of the selected journal.

The rather subjective evaluation techniques, not well described threshold and parameter selection and the lack of mention about some possible annual variability can be considered as shortcomings of the article.

> We address doubts raised by the Reviewer below in responses to detailed comments, especially regarding lines: 169, 193, and 228 (depending on the seasons) and 439 (subjective evaluation).

I recommend accepting the manuscript after minor revision.

Specific comments:

Line 62 - Maybe indicate, that the polarimetric radars are depicted by red circles in the map.

> We have added this information to the caption of Fig. 1 (line 59).

Line 136 – How were the "ground-truth" classes determined?

> The algorithm for detection of convection used in this work is described by Jurczyk et al. (2012) and more briefly by Ośródka and Szturc (2015). We have added this information and reference to the literature.

Line 169 – Does the membership functions depend on the time of year (summer vs. winter cases)?

> We have not differentiated membership functions according to the season of the year. While optimizing this algorithm (DP.TURBINE) on various data, we did not find that its effectiveness significantly depended on the season. We are going to investigate this problem in the further works and to introduce proper complements.

Line 174 – How were the weights obtained – some evaluation algorithm or expert knowledge?

> Weights were optimized by assessing the effectiveness of the DP.TURBINE algorithm on selected data sets which was done visually.

Line 193 – Does the effectiveness depend on the time of year?

> As we wrote above regarding the comment on line 169, while optimizing and validating individual algorithms, we did not find that they required separate optimization depending on the season. We are going to investigate this problem in the further works.

Line 199 – Consider replacing "a lot of" by "several" or some similar term.

We have replaced it.

Line 205 – Maybe a picture without the SPIKE algorithm should be more appropriate.

We generated Fig. 5, right, with the DP.TURBINE / INTERP algorithm only, without using the SPIKE algorithm. It can be noticed that DP.TURBINE weakens the "spike" echoes, which is a side effect: algorithms dedicated to remove non-meteorological echoes are usually effective not only for specific types of echoes, but to a different extent for other non-meteorological echoes. We have added a comment on this to the text describing Fig. 5.

Line 214, 255, 287 – QI chosen by expert knowledge? Why not lower?

Yes, we have determined all QI values based on many years of experience in this matter. The quantitative effect of various disturbances on a specific QI value is a complex issue, as it is practically impossible to determine it in a completely objective manner. This influence depends on the intensity of these echoes as well as on local conditions. Nevertheless, we had to take responsibility for their determination.

We understand that the Reviewer suggests that the QI values we have determined are too high. However, the QI value after corrections due to a specific quality factor should be synchronized with the QI assigned to other types of disturbances, i.e. these values should be consistent: a similar influence of a given factor (e.g. spike or turbines) on data quality should result in a similar decrease in QI. Such estimates are unfortunately largely subjective.

We must bear in mind that these parameters, like the vast majority of parameters introduced in our work, are determined empirically. Therefore, they can only be used for those radars for which they have been determined. For readers who work with data from other radars, the values provided by us are only indicative.

In conclusion: indeed, we did not actually write about these limitations, so now we have added this comment explicitly to line 102 (in the original version). Thank you for drawing our attention to these issues!

Line 220 – Why was sd(Zdr) excluded?

We checked the correlation between the individual dual-pol parameters and the occurrence of non-meteorological echoes, and it turned out that the sd(Zdr) parameter poorly correlated. These results are not included in the article, which is already extensive, so now we have added a brief explanation. Thank you for pointing this out!

Line 228 – Does the threshold values depend on the time of the year (snow vs. liquid precipitation)?

As in the cases on lines 169 and 193, we did not notice the influence of the seasons or the type of precipitation on the parameters of the algorithms for detecting various types of disturbances. We are going to investigate this problem in the further works.

Line 223 – Fig. 6 vs. Fig. 4 - inconsistency about sd(Phidp) – In Fig. 4 is everything below cca. 22° considered as meteorological, in Fig. 6 the threshold is set to 10°. Is there a reason to use different thresholds in the two algorithm?

The membership functions shown in Fig. 4 (for the DP.TURBINE algorithm) resulted from the histograms shown in Fig. 2, with slight modification as a result of verification of the algorithm's performance. We have defined these functions as shown in Fig. 4 in such a way that the histogram from Fig. 2 shows an approximate supremacy in the occurrence of meteorological echoes in the appropriate sd(PhiDP) range, because we assume that the full fuzzy logic algorithm along with the other parameters used here will give us a balanced indication of the type of echo.

The situation is slightly different in Fig. 6 (for DP.NMET), where we used the decision tree, where we apply this parameter more rigorously: this threshold is applied only to echoes with small Rho(HV).

Line 216 – It seems to me from the paper, that the DP.TURBINE was not intentionally designed to detect wind turbines, it just turned out to be better at this type of obstacles, as it is stated later in the line 247.

Yes, you are right! Moreover, the DP.NMET algorithm is an adaptation of the algorithm developed by Seo et al. (2015), which was designed to detect wind turbines (and it was designed for S-band radars of the NEXRAD network). After testing this algorithm on the C-band radars of the Polish network, it turned out to be effective also in the detection of non-meteorological echoes of various origins, while our DP.TURBINE algorithm, which we originally planned to use for various non-meteorological echoes, turned out to be more effective for wind turbine echoes. Currently, in RADVOL-QC, these two algorithms complement each other quite well and respond to both groups of echoes, but to a different extent.

Line 271 – Fig 8. Text labels in the picture are duplicated.

Indeed - we did not notice it… We have corrected this Figure and thank you for pointing this out to us!

Line 439 – The verification of DP.NMET, TURBINE and SPIKE is presented only in forms of some case studies – no statistical evaluation is present.

The numerical verification of particular algorithms is a serious problem because we have no reference, so indirect methods have to be used. In the previous article from 2014, we introduced two coefficients: "symmetry" and "smoothness", for which sums of precipitation must be determined separately for each radar, long enough to be independent of the influence of local convective precipitation, at least monthly. We then argue that the more homogeneous the precipitation field, the better the radar data is corrected. However, it should be remembered that the impact of few types of disturbances can be diagnosed in this way - only those associated with a permanent location of obstacles (groundclutter, wind farms, etc.) excluding disturbances distributed randomly. So we decided not to present this kind of verification. In our opinion, more is the assessment of the effectiveness of correction algorithms on instantaneous measurements, but due to the aforementioned lack of reference (benchmark), we do not see the possibility of a really objective verification.

Section 5 – As the aim of the paper is to present the newly developed and enhanced filtering algorithms, it can be interesting to compare the results in Fig. 12 and 13 where RADVOL-QC is used with and without the new algorithms to see the effect of the presented methods.

Fig. 12: The 2014 version is added.

Fig. 13: After considering this issue, we decided not to add the results obtained for the version of RADVOL-QC from 2014 to Fig. 13. We added a sample radar image to Fig. 12, because indeed, it will be useful to see what progress we have made. However in this case images generated by RADVOL-QC from 2014 would not add anything new to the evaluation of the current version of RADVOL-QC as the performance of the old version in terms of this kind of verification is rather weak - the numbers of exceedances of precipitation 1 mm/10 min are more similar to the ones obtained for the raw data.

Below we have compiled three pictures for an example for the Pastewnik radar - from the left: raw and after the new RADVOL-QC, and at the bottom we added an image after processing with the 2014 version. We can see that the RADVOL-QC from 2014 was not very effective in removal of permanent non-precipitation echoes:

Pastewnik radar

[Figure]

RADVIOL-QC from 2014: